# Socio-Economic Position, Cancer Incidence and Stage at Diagnosis: A Nationwide Cohort Study in Belgium

**DOI:** 10.3390/cancers13050933

**Published:** 2021-02-24

**Authors:** Michael Rosskamp, Julie Verbeeck, Sylvie Gadeyne, Freija Verdoodt, Harlinde De Schutter

**Affiliations:** 1Belgian Cancer Registry, Rue Royale 215, B-1210 Brussels, Belgium; Julie.Verbeeck@registreducancer.org (J.V.); Freija.Verdoodt@kankerregister.org (F.V.); Harlinde.DeSchutter@kankerregister.org (H.D.S.); 2Sociology Department, Interface Demography, Vrije Universiteit Brussel, Pleinlaan 5, B-1050 Brussels, Belgium; Sylvie.Gadeyne@vub.be

**Keywords:** cancer incidence, socio-economic position, cancer stage, nationwide cohort study, census

## Abstract

**Simple Summary:**

Socio-economic position is associated with cancer incidence, survival, and mortality. However, this relationship differs across cancer types and socio-economic parameters. Linking the 2001 census to the nationwide Belgian Cancer Registry for cancer diagnoses between 2004 and 2013, the aim of our study was to evaluate and characterise the associations between individual socio-economic and -demographic indicators and the risk of being diagnosed with six cancer types at the population level. With cancer stage being a major determinant for prognosis, a second aim of this study was to explore whether socio-economic position is associated with stage availability and distribution. This study, encompassing almost seven million individuals, identified population groups at increased risk of cancer and unknown or advanced stage at diagnosis in Belgium, and it contributes to building a more comprehensive picture of the complex and multifaceted nature of socio-economic position and cancer incidence in Belgium.

**Abstract:**

*Background*: Socio-economic position is associated with cancer incidence, but the direction and magnitude of this relationship differs across cancer types, geographical regions, and socio-economic parameters. In this nationwide cohort study, we evaluated the association between different individual-level socio-economic and -demographic factors, cancer incidence, and stage at diagnosis in Belgium. *Methods*: The 2001 census was linked to the nationwide Belgian Cancer Registry for cancer diagnoses between 2004 and 2013. Socio-economic parameters included education level, household composition, and housing conditions. Incidence rate ratios were assessed through Poisson regression models. Stage-specific analyses were conducted through logistic regression models. *Results*: Deprived groups showed higher risks for lung cancer and head and neck cancers, whereas an inverse relation was observed for malignant melanoma and female breast cancer. Typically, associations were more pronounced in men than in women. A lower socio-economic position was associated with reduced chances of being diagnosed with known or early stage at diagnosis; the strongest disparities were found for male lung cancer and female breast cancer. *Conclusions*: This study identified population groups at increased risk of cancer and unknown or advanced stage at diagnosis in Belgium. Further investigation is needed to build a comprehensive picture of socio-economic inequality in cancer incidence.

## 1. Introduction

Cancer is one of the leading causes of death worldwide. While survival has considerably improved for most cancer types over the last few decades, cancer incidence continues to rise around the globe [1]. Several studies have shown a mostly consistent pattern between socio-economic position (SEP) and overall cancer occurrence, reflecting differences in the exposure to main risk factors and inequalities in access to prevention and early detection measures, with a consequent impact on survival and quality of life [2,3]. However, the direction and magnitude of this dynamic and multifaceted relationship differ across cancer types, and the mechanisms by which SEP impacts cancer risk are multiple, diverse, and interconnected [4,5,6]. Behavioural and contextual determinants are the underlying causes behind socio-economic inequality, and no single indicator can capture the complexity of socio-economic (SE) and socio-demographic (SD) circumstances [7,8,9].

The extent of disease at the time of diagnosis is one of the most important determinants for prognosis. After being diagnosed at an early stage, cancer patients can benefit from timely and cure-intended treatment, improving chances of survival but also reducing severe late effects of the cancer-directed treatment [10,11]. However, stage availability and distribution do not seem to be equally dispersed among SE groups [12,13,14,15].

In Belgium, so far, most studies have focused on the impact of SEP on cancer mortality [16,17,18,19]. However, mortality results from incidence and survival, both important epidemiological measures of cancer burden [20]. For Belgium, our previous study indicated an SE gradient in cancer survival for all studied cancer types [21]. Furthermore, Dalton et al. demonstrated that the inequality in cancer survival has increased over time [22].

The purpose of the current study was to evaluate and characterise, for the first time in Belgium, the association between individual-level SE or SD indicators and the risk of being diagnosed with six different cancer types on a nationwide dataset (i.e., lung cancer, colon cancer, rectal cancer, head and neck cancers, breast cancer, and malignant melanoma) while taking demographic and health-related information into account. Second, this study aimed to explore whether SEP is associated with disparities in stage availability and distribution.

## 2. Results

### 2.1. Socio-Economic Position and Cancer Incidence

The study cohort consisted of 6,960,371 individuals (3,338,415 men and 3,621,956 women) aged 25 years or more at time of census (1 October 2001), all of who entered the study on 1 January 2004. During the study period (63,828,547.3 person–years), 280,019 (4.0%) individuals were diagnosed with at least one of the six studied cancer types (286,220 diagnoses), 1,020,999 (14.7%) died, 146,648 (2.1%) were lost to follow-up, and 5,792,724 (83.2%) were still alive at the end of the observation period (31 December 2013; Table 1). 

The number of cancer diagnoses ranged from 7311 malignant melanoma to 54,159 lung cancer cases in men (Table 2a) and from 4338 head and neck cancers to 97,650 breast cancer cases in women (Table 2b). For most cancer types, truncated, age-standardized incidence rates (ASRs) were inversely associated with education and housing comfort. Furthermore, individuals living in couple-based households were at lower risk compared to those living in single-parent, single-person and other types of households. The exceptions were female breast cancer and malignant melanoma patients, where higher incidence rates were observed for individuals with higher education (female breast cancer: from ASR_Primary or lower_ = 253.8 [248.7–259.0] to ASR_Tertiary_ = 330.2 [324.9–335.4]; malignant melanoma: from ASR_Primary or lower_ = 15.5 [14.0–17.0] to ASR_Tertiary_ = 32.8 [31.4–34.2] in men), those living in couple-based households (malignant melanoma: ASR_Couple with(out) child(ren)_ = 23.5 [22.9–24.2] in men; ASR_Couple with(out) child(ren)_ = 31.3 [30.5–32.0] in women), and with higher housing comfort (female breast cancer: from ASR_Tenant and low comfort_ = 257.6 [253.0–262.2] to ASR_Owner and high comfort_ = 289.9 [287.1–292.7]). Lung cancer and head and neck cancers showed the strongest associations (lung cancer: from ASR_Tertiary_ = 91.5 [89.0–94.0] to ASR_Primary or lower_ = 216.4 [212.5–220.3] in men; head and neck cancers: from ASR_Owner and high comfort_ = 28.4 [27.5–29.2] to ASR_Tenant and low comfort_ = 87.8 [84.8–90.8] in men). With the exception of malignant melanoma, the relationship between the SE and SD parameters and incidence rate was more pronounced in men than in women.

Incidence rate ratios (IRRs) were adjusted for age, region of residence, and self-reported health status in multivariable models. Adjusted IRR estimates by SE or SD parameters, cancer type, and sex are shown in Figure 1 and Appendix A.

Lower education was significantly associated with higher incidences of lung cancer, rectal cancer, and head and neck cancers in men and women, with the strongest association (and a tendency towards a gradually increasing IRR with decreasing level of education) observed for lung cancer in men (from IRR_Upper secondary_ = 1.41 [1.36–1.46] to IRR_Primary or lower_ = 2.00 [1.93–2.06]) and women. However, lower education was also associated with a lower incidence for malignant melanoma in men (from IRR_Upper secondary_ = 0.78 [0.73–0.83] to IRR_Primary or lower_ = 0.56 [0.51–0.61]) and women, as well as female breast cancer (from IRR_Upper secondary_ = 0.91 [0.89–0.93] to IRR_Primary or lower_ = 0.78 [0.76–0.80]).

Household composition was associated with the incidence of some cancer types, mainly head and neck cancers and, to a lesser extent, lung cancer. Overall, individuals living alone (single-person households) were at the highest risk for different cancer types, as compared to couples. The strongest associations were observed for head and neck cancers (IRR_Single-person_ = 1.48 [1.42–1.54] in men; IRR_Single-person_ = 1.39 [1.29–1.51] in women). An association was also found for colon, rectal, and breast cancer in women. Furthermore, single-parents showed increased incidences for head and neck cancers and lung cancer in both sexes.

In contrast, malignant melanoma incidences were lower in male single-person households (IRR_Single-person_ = 0.92 [0.85–0.99]) and for single-parents in both sexes (IRR_Single-parent_ = 0.86 [0.75–0.98] in men; IRR_Single-parent_ = 0.91 [0.84–0.98] in women).

House ownership and comfort were also associated with cancer occurrence, mainly for head and neck cancers and lung cancer, for which lower housing conditions were gradually associated with higher incidences for both sexes (head and neck cancers: IRR_Tenant and low comfort_ = 2.35 [2.23–2.46] in men; lung cancer: IRR_Tenant and low comfort_ = 1.75 [1.68–1.83] in women). While a modest tendency (lower housing conditions associated with higher incidences) was observed for colon and rectal cancer in men, this was not the case in women. To some extent, poorer housing conditions were associated with lower IRRs for malignant melanoma and female breast cancer.

Crude incidence models, including one SE or SD parameter per model and only adjusted for age, resulted in slightly more pronounced associations compared to the estimates obtained in the full adjusted models. This was particularly true for lung cancer and head and neck cancers (Appendix A).

### 2.2. Socio-Economic Position and Stage Distribution

Cancer stage was available in 86.4% of cases, ranging from 92.6% in female breast cancer to 77.2% in men lung cancer (Appendix A). The proportion of advanced stage (stage III or IV) diagnoses varied substantially by cancer type: from 6.6% for malignant melanoma in women to 55.9% for lung cancer in men. Furthermore, data availability on stage among elderly patients (aged 65 years or older at diagnosis) was statistically significantly lower compared to younger patients across all studied cancer types.

Both stage availability (Figure 2 and Appendix A) and stage at diagnosis (Figure 3 and Appendix A), reported as adjusted odds ratios (ORs), were related to SE and SD parameters, but the magnitude of association differed largely by cancer type and sex. Figure 2 shows that male lung cancer patients with lower education were more likely to be diagnosed with an unknown stage (OR_Primary or lower_ = 1.14 [1.05–1.23]) compared to those with tertiary education. In women, similar observations were made for head and neck cancer (OR_Lower secondary_ = 1.35 [1.01–1.80]) and breast cancer patients (OR_Primary or lower_ = 1.23 [1.12–1.34]). Living alone was associated with a higher risk of being diagnosed with an unknown stage for lung (OR_Single-person_ = 1.10 [1.04–1.16]) and rectal cancer (OR_Single-person_ = 1.20 [1.03–1.40]) in male patients, as well as for malignant melanoma (OR_Single-person_ = 1.18 [1.01–1.36]) and breast cancer (OR_Single-person_ = 1.16 [1.09–1.24]) in female patients. Single fathers diagnosed with lung cancer were characterised by a higher risk of unknown stage at diagnosis (OR_Single-parent_ = 1.20 [1.07–1.34]). Patients living in ‘other household types’ presented a higher risk of unknown stage at diagnosis for lung and rectal cancer in both sexes and breast cancer in women compared to patients living with a partner. Housing condition was associated with a missing stage for lung (OR_Owner and low comfort_ = 1.09 [1.03–1.15]) and colon cancer (OR_Owner and low comfort_ = 1.14 [1.04–1.25]) in men and for breast cancer (OR_Owner and low comfort_ = 1.10 [1.03–1.17]) in women.

Figure 3 illustrates that lower education was associated with more advanced stage at diagnosis for malignant melanoma (OR_Primary or lower_ = 1.91 [1.45–2.51]) and lung cancer (OR_Primary or lower_ = 1.09 [1.01–1.18]) in males and for breast cancer (OR_Primary or lower_ = 1.22 [1.15–1.29]) in females. Household composition was also related with a higher risk of advanced-stage diagnosis for rectal cancer (OR_Single-parent_ = 1.48 [1.20–1.83]), head and neck cancers (OR_Single-person_ = 1.22 [1.10–1.35]), and malignant melanoma (OR_Single-person_ = 1.33 [1.05–1.68]) in men, as well as for female breast cancer with individuals living in single-person (OR_Single-parent_ = 1.07 [1.02–1.12]), single-parent (OR_Single-person_ = 1.10 [1.04–1.17]) and other types of households (OR_Other household_ = 1.17 [1.07–1.29]), showing higher odds ratios for advanced-stage diagnosis. In contrast, single female colon cancer patients seemed to be diagnosed more often in early-stage disease compared to couples (OR_Single-person_ = 0.93 [0.87–0.99]). Lower housing quality was associated with higher risk of advanced-stage rectal cancer (OR_Tenant and low comfort_ = 1.18 [1.04–1.33]) and head and neck cancers (OR_Tenant and low comfort_ = 1.45 [1.29–1.63]) in men and advanced-stage lung cancer (OR_Tenant and low comfort_ = 1.21 [1.08–1.34]), head and neck cancers (OR_Tenant and low comfort_ = 1.42 [1.15–1.76], breast cancer OR_Tenant and low comfort_ = 1.32 [1.25–1.40]), and malignant melanoma (OR_Tenant and low comfort_ = 1.88 [1.44–2.46]) in women.

Estimates from the crude models for stage availability and stage at diagnosis, including only one SE or SD parameter per model and only adjusted for age, differed only slightly compared to those obtained in the full adjusted models. The association between SE or SD parameters and stage at diagnosis in the crude models was slightly more pronounced for head and neck cancers and malignant melanoma compared to the full adjusted models (Appendix A).

### 2.3. Sensitivity Analyses

To assess potential mediator effects, we omitted self-reported health status and region of residence as covariates, which did not result in substantial changes in IRR estimates. While there was a tendency towards a slightly more pronounced association between the distinct SE or SD parameters and outcome for some cancer types, this relationship was not consistently observed through all cancer types (Appendix A).

Similar observations were made for stage availability and distribution: no substantial effect was observed when removing self-reported health and region of residence as adjustment factors (Appendix A).

## 3. Discussion

This large cohort study encompassing almost seven million individuals provides evidence of disparities in cancer incidence and stage at diagnosis according to individual SEP at the Belgian population level for the period of 2004–2013. Adding to the growing body of evidence, it also provides new and important findings in this yet understudied subject on a nationwide scale in the light of the Belgian health care system. The study contributes towards a more comprehensive overview of SE inequality in cancer burden in Belgium, a country known for its obliged and advanced health insurance.

SEP is a theoretical construct, conceptualized through indicators for which assessment can vary in the number of included SE/SD components and the level of measurement. Considering individual SE or SD variables may uncover the complex, multifaceted, and dynamic nature of deprivation and its influence over the life course. Measures of SEP at the individual level are not interchangeable and reflect different aspects of deprivation through material, cognitive, cultural, and (psycho)social aspects. In this study, we focused on multiple SE and SD parameters at the individual level, namely education level, household composition, and housing conditions. Reflecting chances in early life, education confers a broad set of resources to access information, acquire knowledge, and critically assess facts. Strongly influenced by that of parents, SEP is typically fixed during adult life and a strong determinant of employment, income, and social networks. In addition, education is also associated with health literacy, behaviour, and symptom perception, which may lead to increased communication with healthcare professionals and shorter delays in seeking health care [23]. Household composition/living arrangement, accounting for non-marital relationships, provides insights into an individual’s social environment. Persons living together/cohabitating might benefit from a healthier lifestyle and stronger social support and might be more encouraged to seek health care in time and comply with health advice [24,25]. While Belgium has a comprehensive and nationwide, largely tax-funded health insurance program where equity in rights and access to healthcare are pursued, financial advantage may affect access to and usage of health care [26]. Housing conditions and ownership relate to material circumstances and accumulated wealth and may have direct health effects through exposure to risk factors [9,27].

In general, we observed increased cancer risk among the most deprived. This was particularly the case for lung and head and neck cancers, with the most pronounced associations for men, in which these cancers are also more frequent [28,29,30]. A higher consumption of alcohol and tobacco among the most deprived groups may play a role but do not, according to literature, entirely explain the observed SE differences [2,31,32,33]. In fact, numerous studies have found that smoking explained the differences in lung cancer incidence across SE groups for roughly 50–70% [34,35,36]. Occupational social class and exposure to occupational carcinogens, as well as air pollution, might play an important role. Further understanding in lung cancer aetiology is needed to clarify the pathways linking SEP to lung cancer. Similar observations were made for head and neck cancers, where tobacco and alcohol consumption, even though representing the main risk factors, could not explain all the differences observed across SE groups [37]. In addition, Krupar et al. showed that prevalence of HPV infection, an independent risk factor for different head and neck cancers, differed across regions and SE factors [38]. Differences in exposure but also in susceptibility to common risk factors across SE groups may have led to the observed disparities in cancer incidence. Furthermore, a lower SEP was associated with an advanced stage at diagnosis for lung and head and neck cancers in men, which might be influenced by fatalistic beliefs and attitudes, differences in health care-seeking behaviour, and a higher prevalence of comorbid conditions related to common risk factors, entailing a possible misinterpretation of alarming symptoms and a delay in presentation at the clinic [39,40,41,42,43,44].

In contrast to the general patterns observed, a higher SEP was associated with an increased risk for malignant melanoma in both sexes. High intermittent exposure to ultra-violet radiation in the more affluent population, typically due to intensive (holidays) sun-bathing and the use of sunbeds, seems the most important risk behaviour [45]. However, studies have also found that the increased risk among the high SE groups is associated with less extensive tumours and a better outcome [46]. This is in line with our study showing that an advanced stage at diagnosis was associated with a lower SEP in both men and women. Persons with a higher SEP are reported to have a better knowledge about the disease and more frequently consult physicians and dermatologists (for marks and moles) leading to earlier diagnoses [47,48].

For female breast cancer, the situation is more complex. International findings support that a higher education level is associated with a higher risk for breast cancer [3,49]. Though enhancing healthy behaviours and uptake of screening, higher education is typically also linked with nulliparity or oligoparity, the postponement of parenthood, and hormone replacement therapy usage [50,51]. These findings suggest that the association between level of education and breast cancer risk could, to a substantial degree, be explained by these established risk factors [52,53,54,55]. However, a lower SEP was strongly associated with a higher risk of advanced stage breast cancer, which is in line with international findings and could reflect social patterning in the awareness of cancer symptoms and seeking medical help in a timely manner [12,14,56].

Regarding colon and rectal cancer, differences in risk according to SEP were less pronounced. Lower education was associated with an increased incidence in rectal cancer in both sexes, and poorer housing conditions were associated with an increased colon and rectal cancer incidence in men. The relationship between colorectal cancer and SEP is not homogeneous between European countries. While engaging in high-risk behaviour may be expected in more deprived populations, greater participation in screening programs among the more affluent influences on colorectal cancer risk can be too [12,30,50,57]. Furthermore, household composition and poorer housing conditions were associated with advanced-stage rectal cancer. These results were in line with international studies indicating that SEP is a predictor of colorectal cancer stage at diagnosis [58,59,60,61].

Some cancer types are hard to suspect because patients present with non-specific symptoms (e.g., chest, back, or abdominal pain). This diagnostic difficulty, potentially enforced by the comorbid conditions of the patient, may lead to multiple consultations in primary care and delay in hospital referral, thus increasing the risk of stage progression. For others cancer types, specific signs and symptoms, such as palpable breast lumps or visible skin lesions, make cancer easier to suspect. In our study, differences in stage distribution across SE groups tended to be more pronounced for these cancer types and likely pointed towards potential delays in medical consultation and differences in the help-seeking behaviour and health literacy of the patients [62,63,64].

Stage availability at diagnosis was affected by the SE and SD parameters, showing fewer known stages amongst most deprived patients, but the magnitude of the association differed largely by cancer type and sex. Male lung cancer patients with lower education were more likely to be diagnosed with unknown stage. In women, similar observations were made for breast and head and neck cancers. Living alone was associated with a higher risk of a missing stage for lung and rectal cancer in men, as well as for malignant melanoma and breast cancer in women, while poor housing conditions were associated with a missing stage for lung and colon cancers in men and breast cancer in women. These results indicated that a missing stage is not randomly distributed among cancer types and patients.

Besides being linked to ethnicity, marital status, and place of residence, missing stage information has also been related to clinical characteristics, such as higher age and frailty, the presence of comorbid conditions, and the need for complex or intense care and institutionalization [14,15,65,66,67]. Reported associations between comorbidity and SEP might at least partially explain higher missing stage proportions in less favourable SE groups [68,69]. Patients with poor prognoses may die before any diagnostic investigation can be conducted, or they may be considered too frail to undergo staging investigations. This is also supported by low survival estimates for patients with a missing stage at diagnosis [70,71]. However, overall stage availability and distribution differed across cancer types. This should be considered when interpreting effect size, which varied across cancer types.

The observed disparities in stage availability likely had an impact on the differences found in stage distribution, and the results require careful interpretation. Stage distribution analyses were performed in a subgroup of the cancer population (i.e., those with a known stage); generalization may not be feasible because our analyses on stage availability indicated a SE gradient for missing stage. Given the passive cancer registration in Belgium relying on notifications provided by oncological cancer programs and laboratories for pathological anatomy, a source-dependent variation in stage availability cannot be excluded. While relevant for the clinical management of patients, qualitative and complete cancer stage registration is also crucial for the understanding of outcome measures at the population level and for assessing the efficacy of early detection programs [72]. Understanding the association between SEP and the availability of stage at diagnosis, as well as the potential underlying mechanisms responsible for the non-random distribution of missing stage data, is crucial for reaching valid conclusions and assessing the potential extent of bias. Furthermore, this understanding may lead towards the implementation of reasonable measures to further increase stage availability.

Our study had several limitations. We could not account for known risk factors (e.g., alcohol and tobacco consumption, body mass index, and HPV status in head and neck cancers), reproductive history (breast cancer), or participation in screening programs (breast and colorectal cancer), and residual confounding cannot be excluded. Data available from a 2001 census was considered to define the individual’s SEP. However, this also induced the assumption that the person’s situation did not change between the census (cross-sectional information) and the end of the observation period. Furthermore, self-reported exposure data could have potentially affected results through subjective perception or social desirability, amongst others [73]. Individual-level SE measures are the best way to explore the underlying mechanisms by which SEP influences cancer risk in depth. While other studies have shown an independent association between both individual- and area-based SEP and cancer incidence, the effect of neighbourhood deprivation is supposed to be relatively modest compared to that of an individual SEP [74]. Given the available data, we chose to focus on three distinct SE indicators. Additional parameters such as income and occupation would have provided a more complete vision of how different aspects of SEP interact with cancer incidence and stage. However, such information was not available at the population level. Less stable over time, this information would be needed on a more regular basis (e.g., yearly). Through sensitivity analyses, we evaluated the potential mediating effect of self-reported health status [75,76,77,78]. While there was a tendency towards a slightly more pronounced association between the distinct SE or SD parameters and outcome for some cancer types, when removing self-reported health from the models, this finding was not consistently observed for all studied cancer types. Furthermore, by including all SE or SD parameters into one model, the aim of our study was to single out the association between each indicator and the outcome. Crude estimates for each SE or SD parameter only adjusted for age were also calculated, resulting in slightly more pronounced associations compared to those obtained in the fully adjusted models. However, this did not result in substantial changes in IRR and OR estimates. Though compulsory, census information on education level and housing conditions was missing for some individuals, with SE inequalities in (health) survey participation being demonstrated in the past [79,80]. Missing data may have biased our results [81]. By creating an additional, fictive category for missing values, we included those observations in our study cohort without making any assumptions on the missingness of the data. Information on education level, housing conditions, and self-reported health was missing in 12.8%, 8.5%, and 5.4% of the study cohort, respectively. The estimates obtained for the category of missing education tended towards those observed for the lowest education category for some cancer types, suggesting that missingness was not randomly distributed. Meanwhile, for missing housing conditions, no clear pattern was observed. However, the focus of this study was to evaluate patterns of association between SEP and outcomes, and the impact of potential bias by missing data was unlikely to be large enough to substantially impact the observed patterns of association. Finally, this study focused on the period from 2004 to 2013, and future work should assess whether the observed disparities in cancer incidence and stage at diagnosis changed over the more recent years in Belgium [82,83,84].

## 4. Material and Methods

### 4.1. Data Sources and Study Cohort

The Belgian 2001 population census [85] was linked to the nationwide Belgian Cancer Registry (BCR) [86] database for incidence years of 2004–2013. Persons aged 25 years or older at time of census (1 October 2001) entered the study cohort at 1 January 2004 and were followed until first cancer diagnosis of interest, emigration (lost to follow-up), death, or 31 December 2013—whichever occurred first. Vital status and emigration information was provided by the Crossroads Bank for Social Security (CBSS) [87]. Database linkage was enabled through the patient’s unique national social security identification number (1:1 linkage) assigned to all residents in Belgium. Individuals younger than 25 years were excluded because they might still have been in the educational system and/or are not settled yet.

Information regarding SE and SD parameters was obtained from the 2001 compulsory questionnaire-based population census. The variables that we evaluated in our study were education level, household composition, and housing conditions. Education level reflects the highest attained degree and was classified into primary or less, lower secondary, upper secondary, and tertiary education. Household composition was categorized into couples with or without children, single-persons, single-parents, and other household types (other private households and collective households/institutions). The variable measuring housing conditions were classified in four categories by combining home ownership (owner vs. tenant) and housing comfort (high vs. low) [18]. Information on education level and housing condition was missing in 12.8% and 8.5% of the study cohort, respectively. These missing values were considered as distinct categories in the analyses.

The population-based BCR includes all incident cancer cases from 2004 onwards, relying on information from the oncological care programs and laboratories for pathological anatomy. Six common cancer types (International Classification of Diseases—10th Edition [88]) were considered: lung cancer (C34), colon cancer (C18–C19), rectum cancer (C20), head and neck cancers (C00–C14 and C30–C32), female breast cancer (C50), and malignant melanoma (C43). Patients could account for multiple diagnoses across cancer types; however, only the first occurring diagnosis was considered within each cancer type. Cancer data included the date of diagnosis and the stage at diagnosis. The validity of stage information depends strongly on the data received from the sources, i.e., the oncological care programs and the laboratories for pathological anatomy. All data that enter the BCR are submitted to an extended set of automated and manual validation procedures based on IARC (International Agency for Research on Cancer) guidelines to ensure the validity and quality of the data. The data source was consulted to provide additional details for cases with insufficient, erroneous, or conflicting information or an uncertain diagnosis. A combined stage was considered for the analyses, meaning that: pathological stage, if known, prevailed over the clinical stage, except for cases with clinical proof of distant metastasis (i.e., TNM cM = 1) [86]. Patients with a missing stage were considered as a distinct category.

The definition of the observation period and study population is provided as Appendix A.

### 4.2. Incidence Analyses and Stage Distribution

Truncated age-standardized incidence rates (ASRs/100,000 person–years) were calculated by SE and SD parameters by considering the age distribution of the total population at start of the study (1 January 2004) as the standard population and the accumulated person–years to calculate age-specific incidence rates in 5-year wide age intervals. As the youngest individuals entering the study were aged 27 years on 1 January 2004, ASRs were left-truncated below age 30 years and computed as the sum of weighted age-specific incidence rates (Appendix A) [89].

Adjusted IRRs were computed through Poisson regression models with the number of diagnoses as the dependent variable. Education level, household composition, and housing conditions were included in every model as independent variables, and age at the start of follow-up, region of residence, and self-reported health at time of census were included as control variables. The log of the person–time was used as offset variable.

Crude estimates for the associations between SE or SD parameters and cancer incidence were assessed by including a single SE or SD parameter per model and only adjusting for age, which was considered a major confounder.

Differences in stage distribution and availability (registration of stage) between SE or SD categories were estimated through logistic regression models and reported as ORs. Two types of analyses were performed: (i) the probability of being diagnosed with an unknown (vs. known) stage and (ii) the probability of being diagnosed with an advanced (III–IV vs. early (I–II)) stage, among cases diagnosed with a known stage. Cases for which staging was not applicable according to the TNM classification rules (TNM 6th and 7th editions) were excluded from these stage-specific analyses. Education level, household composition, and housing conditions were included in every model, and adjustment factors included age and region of residence at time of diagnosis, as well as self-reported health at time of census. To assess the potential mediating effects of SE or SD parameters on each other, the crude models included a single SE or SD parameter per model and were only adjusted for age, which was considered a major confounder.

Reference categories for IRR and stage-specific analyses included the presumably most advantaged groups, i.e., individuals with tertiary education, living in couple-based households, and owners of high comfort housings.

All the analyses were stratified by sex and cancer type. Tests for statistical significance were 2-sided at a α = 0.05 level of significance and 95% confidence intervals (95% CI). SAS 9.4 (SAS Institute Inc., Cary, NC, USA) was used to conduct the analyses and to create the figures.

### 4.3. Sensitivity Analyses

To assess the potential mediator effects of self-reported health and region of residence on cancer incidence, stage availability and stage at diagnosis, adjusted estimates were calculated without considering those adjustments factors.

## 5. Conclusions

This study highlights the association of distinct SE and SD factors at the individual level with cancer risk and stage at diagnosis. SE inequalities in cancer are an important public health issue. An improved understanding of the interplay between SEP and cancer risk increases possibilities to launch targeted interventions to reduce the SE gradient. Earlier detection and prevention measures should especially focus on individuals being at higher risk of advanced stage diagnosis.

## Figures and Tables

**Figure 1 cancers-13-00933-f001:**
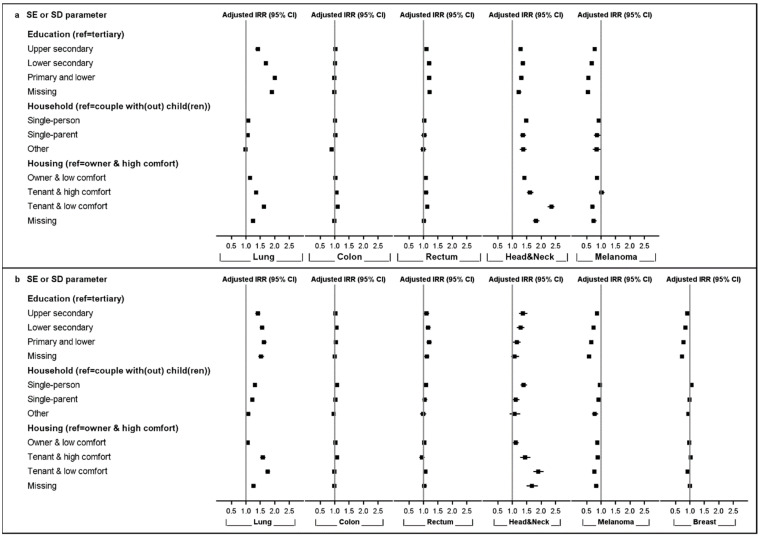
Adjusted incidence rate ratios (IRRs) with a 95% confidence interval (CI) by socio-economic (SE) or -demographic (SD) parameters, cancer type, and sex; 2004–2013. Upper panel (**a**): men; lower panel (**b**): women.

**Figure 2 cancers-13-00933-f002:**
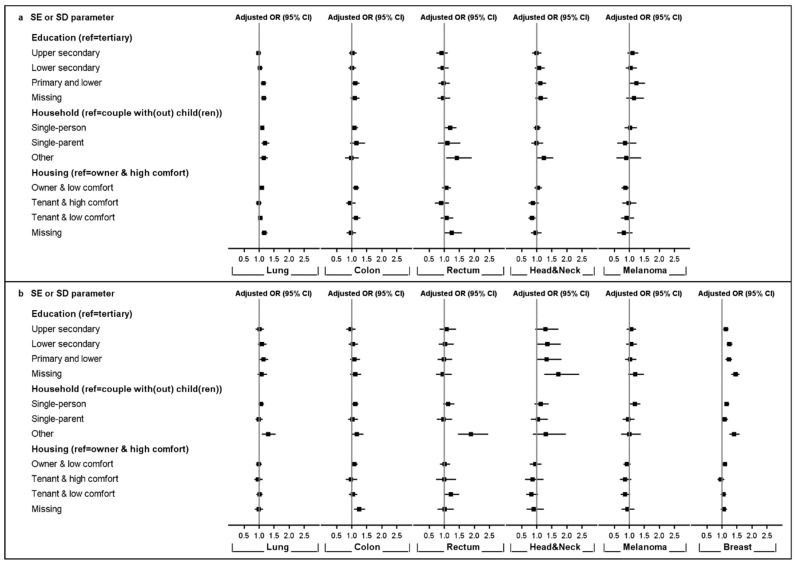
Adjusted odds ratios (ORs) with a 95% confidence interval (CI) for being diagnosed with unknown stage by socio-economic (SE) or -demographic (SD) parameters, cancer type, and sex; 2004–2013. Upper panel (**a**): men; lower panel (**b**): women.

**Figure 3 cancers-13-00933-f003:**
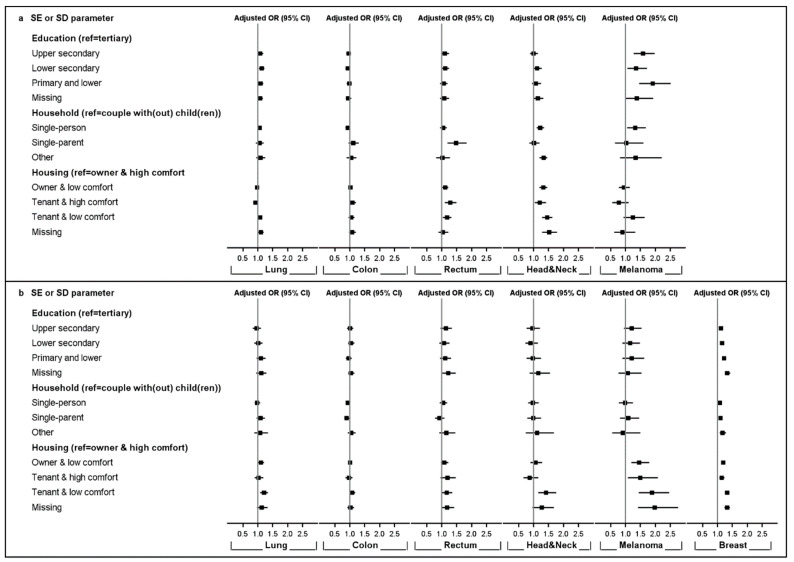
Adjusted odds ratios (ORs) with a 95% confidence interval (CI) for being diagnosed with advanced stage among known stages by socio-economic (SE) or -demographic (SD) parameters, cancer type, and sex; 2004–2013. Upper panel (**a**): men; lower panel (**b**): women.

**Table 1 cancers-13-00933-t001:** Socio-economic and -demographic characteristics of the study population at baseline (1 January 2004), based on census 2001 data.

Socio-Economic or -Demographic Factor	Men				Women				Overall	
	N	%	Age (Mean)	Age (SD)	N	%	Age (Mean)	Age (SD)	N	%
Overall	3,338,415	100.0	51.5	15.3	3,621,956	100.0	54.1	16.8	6,960,371	100.0
Education Level										
Tertiary	485,656	14.5	47.1	13.6	661,712	18.3	44.7	13.2	1,147,368	16.5
Upper secondary	775,460	23.2	46.2	13.7	822,717	22.7	47.2	14.3	1,598,177	23.0
Lower secondary	873,946	26.2	52.2	14.2	853,279	23.6	55.8	15.0	1,727,225	24.8
Primary or lower	789,766	23.7	63.0	13.9	810,116	22.4	66.9	14.2	1,599,882	23.0
Missing	413,587	12.4	56.4	16.3	474,132	13.1	61.8	17.3	887,719	12.8
Household Composition										
Couple with(out) child(ren)	2,546,249	76.3	51.7	15.0	2,448,876	67.6	54.1	16.8	4,995,125	71.8
Single-parent household	540,808	16.2	46.8	13.4	666,317	18.4	49.5	14.9	1,207,125	17.3
Single-person household	143,615	4.3	51.2	16.4	346,502	9.6	64.7	17.9	490,117	7.0
Other	107,743	3.2	53.9	18.8	160,261	4.4	69.5	19.9	268,004	3.9
Housing Conditions										
Owner and high comfort	1,459,736	43.7	51.8	14.2	1,482,715	40.9	51.6	14.3	2,942,451	42.3
Owner and low comfort	234,608	7.0	53.8	16.1	250,336	6.9	58.0	17.4	484,944	7.0
Tenant and high comfort	895,643	26.8	48.2	14.9	1,025,794	28.3	48.1	14.9	1,921,437	27.6
Tenant and low comfort	471,506	14.1	47.0	15.5	546,887	15.1	52.8	18.3	1,018,393	14.6
Missing	276,922	8.3	51.8	17.0	316,224	8.7	60.2	20.0	593,146	8.5
Self-Reported Health										
Good	2,279,474	68.3	48.1	14.0	2,304,880	63.6	49.1	14.8	4,584,354	65.9
Poor	879,249	26.3	60.6	15.0	1,123,099	31.0	64.1	15.8	2,002,348	28.8
Missing	179,692	5.4	49.9	15.6	193,977	5.4	56.5	18.6	373,669	5.4
Region of Residence										
Brussels Capital Region	297,828	8.9	49.9	15.8	339,249	9.4	53.6	17.8	637,077	9.2
Flemish Region	1,988,143	59.6	51.9	15.3	2,108,807	58.2	54.1	16.7	4,096,950	58.9
Walloon Region	1,052,444	31.5	51.3	15.2	1,173,900	32.4	54.3	16.8	2,226,344	32.0

SD: Standard deviation.

**Table 2 cancers-13-00933-t002:** Number of diagnoses (N) and left-truncated (30+ years) age-standardized incidence rates (ASRs) with a 95% confidence interval (CI) by socio-economic and -demographic factors and cancer type among men (a) and women (b) over 2004–2013.

**(a). Number of diagnoses (N) and left-truncated (30+ years) age-standardized incidence rates (ASRs) with a 95% confidence interval (CI) by socio-economic and -demographic factors and cancer type among men over 2004–2013.**																			
**Socio-Economic or -Demographic Factor**	**Lung Cancer**					**Colon Cancer**				**Rectal Cancer**				**Head and Neck Cancers**				**Malignant Melanoma**	
	**N**		**ASR [95% CI]**			**N**	**ASR [95% CI]**			**N**		**ASR [95% CI]**		**N**		**ASR [95% CI]**		**N**	**ASR [95% CI]**
Overall	54,159		160.55 [159.2–161.9]			29,067	85.4 [84.4–86.4]			13,839		40.8 [40.1–41.5]		14,824		44.1 [43.4–44.8]		7311	22.5 [22.2–23.1]
Educational Degree																			
Tertiary	5525		91.5 [89.0–94.0]			4969	83.1 [80.7–85.5]			2218		34.7 [33.2–36.2]		2100		30.4 [29.0–31.7]		2259	32.8 [31.4–34.2]
Upper secondary	8375		133.8 [130.8–136.8]			5322	88.2 [85.7–90.6]			2521		39.4 [37.8–41.0]		3062		41.6 [40.0–43.1]		1800	24.6 [23.4–25.8]
Lower secondary	13,709		166.0 [163.2–168.8]			6995	86.4 [84.4–88.5]			3550		43.2 [41.8–44.6]		4137		48.3 [46.8–49.7]		1598	20.3 [19.3–21.3]
Primary or lower	16,281		216.4 [212.5–220.3]			7331	86.0 [83.5–88.5]			3437		43.2 [41.6–44.9]		3075		56.8 [54.3–59.2]		989	15.5 [14.0–17.0]
Missing	10,269		212.1 [207.8–216.4]			4450	84.3 [81.4–86.9]			2113		41.8 [39.9–43.7]		2450		59.6 [57.2–62.0]		665	15.5 [14.2–16.8]
Household Composition																			
Couple with(out) child(ren)	41,246		152.9 [151.5–154.4]			23,055	85.2 [84.1–86.4]			10,950		40.6 [39.8–41.4]		9806		36.7 [36.0–37.5]		5989	23.5 [22.9–24.2]
Single-person household	9439		197.1 [193.1–201.1]			4386	87.7 [85.0–90.4]			2056		41.8 [40.0–43.7]		3655		76.4 [73.9–78.9]		936	19.1 [17.8–20.3]
Single-parent household	1778		172.1 [163.1–181.0]			805	86.4 [79.8–93.0]			426		42.4 [37.9–46.9]		795		58.5 [54.0–63.0]		228	20.0 [17.1–22.9]
Other	1696		182.2 [173.3–191.1]			821	77.7 [72.1–83.3]			407		40.8 [36.6–44.9]		568		67.3 [61.7–72.9]		158	17.2 [14.5–20.0]
Housing Conditions																			
Owner and high comfort	20,152		127.2 [125.4–129.0]			13,101	83.6 [82.2–85.0]			6153		38.8 [37.8–39.8]		4575		28.4 [27.5–29.2]		3957	26.4 [25.5–27.2]
Owner and low comfort	16,790		169.9 [167.3–172.6]			8872	86.7 [84.8–88.5]			4283		42.9 [41.6–44.2]		4254		46.4 [45.0–47.8]		1776	19.7 [18.8–20.7]
Tenant and high comfort	3477		179.5 [173.5–185.5]			1733	90.1 [85.7–94.3]			825		42.1 [39.2–45.0]		1014		49.4 [46.3–52.5]		542	26.0 [24.0–28.5]
Tenant and low comfort	8939		249.7 [244.4–254.9]			3268	91.7 [88.5–94.8]			1598		44.2 [42.0–46.4]		3353		87.8 [84.8–90.8]		638	16.1 [14.8–17.3]
Missing	4801		200.1 [194.3–205.8]			2093	82.3 [78.7–85.9]			980		39.4 [36.8–41.9]		1628		69.7 [66.3–73.1]		398	16.6 [15.0–18.3]
**(b). Number of diagnosis (N) and left-truncated (30+ years) age-standardized incidence rates (ASRs) with 95% confidence interval (CI) by socio-economic and -demographic factors and cancer type, among women over 2004–2013.**																			
**Socio-Economic or -Demographic Factor**	**Lung Cancer**			**Colon Cancer**				**Rectal Cancer**			**Head and Neck Cancers**				**Malignant Melanoma**			**Breast Cancer**	
	**N**	**ASR [95% CI]**		**N**	**ASR [95% CI]**			**N**	**ASR [95% CI]**		**N**		**ASR [95% CI]**		**N**		**ASR [95% CI]**	**N**	**ASR [95% CI]**
Overall	19,493	54.5 [53.7–55.3]		26,157	72.2 [71.3–73.1]			9137	25.3 [24.8–25.8]		4338		12.0 [11.6–12.4]		10,084		26.7 [29.1–30.3]	97,650	279.0 [277.2–280.8]
Educational Degree																			
Tertiary	2035	40.3 [38.3–42.4]		3005	69.7 [66.9–72.6]			1105	23.0 [21.4–24.6]		571		9.9 [8.9–10.8]		2830		39.2 [37.5–40.9]	21,380	330.2 [324.9–335.4]
Upper secondary	3500	52.7 [50.8–54.6]		3966	71.9 [69.5–74.2]			1512	24.8 [23.5–26.2]		952		13.3 [12.3–14.2]		2535		32.5 [31.1–33.8]	21,802	297.3 [293.1–301.6]
Lower secondary	5481	58.9 [57.3–60.5]		6614	74.1 [72.3–75.9]			2362	26.0 [24.9–27.1]		1222		13.1 [12.3–13.8]		2207		26.6 [25.4–27.9]	24,215	273.5 [270.0–277.1]
Primary or lower	5217	68.7 [66.2–71.1]		8163	73.5 [71.5–75.4]			2714	27.1 [25.7–28.5]		920		13.1 [12.0–14.2]		1611		20.5 [18.8–22.2]	18,963	253.8 [248.7–259.0]
Missing	3260	65.0 [62.6–67.4]		4409	69.0 [66.8–71.1]			1444	24.6 [23.2–26.0]		673		14.6 [13.5–15.8]		901		18.4 [17.1–19.8]	11,290	233.9 [229.2–238.6]
Household Composition																			
Couple with(out) child(ren)	11,621	48.5 [47.6–49.4]		15,038	71.2 [70.0–72.3]			5528	24.7 [24.0–25.4]		2549		10.3 [9.9–10.7]		7278		31.3 [30.5–32.0]	66,882	279.5 [277.4–281.7]
Single-person household	5144	75.3 [73.0–77.7]		7874	74.7 [72.8–76.7]			2484	26.2 [25.0–27.5]		1153		19.2 [17.9–20.4]		1715		28.8 [27.2–30.5]	18,861	288.4 [283.5–293.4]
Single-parent household	2025	66.4 [63.3–69.6]		1800	72.9 [69.4–76.5]			679	25.8 [23.7–27.9]		477		14.6 [13.2–16.1]		819		25.1 [23.2–26.9]	8651	271.2 [265.0–277.5]
Other	703	58.8 [53.8–63.9]		1445	69.6 [65.0–74.1]			446	25.3 [22.2–28.2]		159		13.2 [10.7–15.7]		272		22.2 [18.9–25.5]	3256	255.6 [244.7–266.5]
Housing Conditions																			
Owner and high comfort	6432	42.9 [41.8–44.0]		9375	71.0 [69.5–72.5]			3462	24.6 [23.7–25.4]		1406		9.2 [8.7–9.7]		4982		34.9 [33.9–36.0]	43,837	289.9 [287.1–292.7]
Owner and low comfort	5464	50.9 [49.6–52.3]		9284	73.6 [72.0–75.1]			3090	25.8 [24.8–26.7]		1153		11.0 [10.3–11.6]		2667		27.3 [26.2–28.4]	27,927	270.8 [267.5–274.1]
Tenant and high comfort	1525	74.1 [70.3–78.0]		1366	77.4 [73.1–81.6]			445	23.4 [21.2–25.7]		307		13.9 [12.3–15.5]		674		29.2 [26.9–31.5]	6498	299.3 [291.7–306.8]
Tenant and low comfort	4375	95.2 [92.3–98.0]		3703	72.0 [69.6–74.4]			1330	26.8 [25.4–28.3]		999		21.7 [20.3–23.0]		1138		22.9 [21.6–24.3]	12,262	257.6 [253.0–262.2]
Missing	1697	64.3 [61.1–67.5]		2429	71.0 [67.9–74.0]			810	25.3 [23.5–27.2]		473		18.4 [16.7–20.2]		623		23.0 21.1–24.9]	7126	263.0 [256.6–269.4]

## Data Availability

The cancer cohort data used and analyzed during the study are available from the corresponding author upon reasonable request. The pseudonymized data can be provided within the secured environment of the Belgian Cancer Registry after having been guaranteed that GDPR regulations will be applied.

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
