# Peer review of "Socio-Economic Position, Cancer Incidence and Stage at Diagnosis: A Nationwide Cohort Study in Belgium"

_cancers, 2021, doi:10.3390/cancers13050933_

Round 1

Reviewer 1 Report

I do not agree in the presentation of the adjusted models. In my opinion (but please consider statistical review) the main model presented in the manuscript should be the model not adjusting for self-reported health, as this is a mediator (at least for educational level). Please include more information about your choice of adjustment and the expected causal pathways. E.g. when educational level is the exposure, the model  should NOT adjust for other SEP-parameters as these can be considered as mediators, and you will (most likely) underestimate the effect of education.

In a sub-analysis, you could adjust for self-reported health in order to investigate, in a simplistic way, the possible indirect effects of self-reported health. And please then discuss how and the mechanisms by which self-reported health mediates the association in question SEP --> cancer incidence / stage at diagnosis.

A general comment: The concept “socioeconomic position” is recommend instead of “socioeconomic status”, consider to revise in the entire manuscript (Galobardes B, Shaw M, Lawlor DA, Lynch JW, Davey Smith G. Indicators of socioeconomic position (part 1). J Epidemiol Community Health. 2006 Jan;60(1):7-12. doi: 10.1136/jech.2004.023531. PMID: 16361448; PMCID: PMC2465546.) And please revise the manuscript once more for the different use of concepts: Socioeconomic status, socioeconomic deprivation ect.

L.18: Revise to: “whether socioeconomic position is associated with stage availability and distribution”?

  1. 22: Revise to: “socioeconomic position and cancer incidence”

  1. 31: Delete “more”

  1. 49, l. 58: revise deprivation to position.

l.61 consider to add.: Further, the inequlity in survival has increased over time. (Dalton SO, Olsen MH, Johansen C, Olsen JH, Andersen KK. Socioeconomic inequality in cancer survival - changes over time. A population-based study, Denmark, 1987-2013. Acta Oncol. 2019 May;58(5):737-744. doi: 10.1080/0284186X.2019.1566772. Epub 2019 Feb 11. PMID: 30741062.)

Author Response

Comments and Suggestions Reviewer 1

I do not agree in the presentation of the adjusted models. In my opinion (but please consider statistical review) the main model presented in the manuscript should be the model not adjusting for self-reported health, as this is a mediator (at least for educational level). Please include more information about your choice of adjustment and the expected causal pathways. E.g. when educational level is the exposure, the model  should NOT adjust for other SEP-parameters as these can be considered as mediators, and you will (most likely) underestimate the effect of education.

In a sub-analysis, you could adjust for self-reported health in order to investigate, in a simplistic way, the possible indirect effects of self-reported health. And please then discuss how and the mechanisms by which self-reported health mediates the association in question SEP --> cancer incidence / stage at diagnosis.

In our main analyses, we included all three SE or SD parameters in one model, since the aim of our study is to single-out the ‘direct’ effect -not necessarily the total effect- of each investigated indicator on cancer incidence and stage availability/distribution. Nonetheless, we agree with the reviewer that household composition and housing conditions may be on the pathway between education and the investigated outcomes. In order to allow the reader to evaluate the potential ‘conservative bias’ of including all SE/SD parameters in the model, we also provide ‘crude’ estimates (only adjusting for age, which is considered a major confounder in the analyses).

In our study, health status was self-reported and refers to an individual and subjective perception at a certain point in time (census 2001). This perceived health might be partly influenced by the individual’s health literacy, which in turn, could be influenced by the individual’s education level. However, the contribution of education level on self-reported health and the pathways linking education level and self-reported health are varying throughout literature. In a broader sense, the potential mediating effect of self-reported health between the different investigated socioeconomic or -demographic parameters and the studied outcomes in our study aren’t yet fully clear [Weenberg et al. 2012]. Cultural, social and demographic aspects next to biological and psychological dimensions can be integrated when (physical and/or mental) health is rated by an individual [Manderbacka et al. 1998]. While self-reported health has been shown to be a strong determinant of mortality for a number of major chronic diseases, its association with cancer etiology is not yet fully clear [Roelsgaard et al. 2016, Jackson et al. 2019]. We are not fully convinced self-reported health is a mediator in our study, however we admit there is a risk of it being a mediator and adjusting for it may have a conservative effect on our estimates.  By removing self-reported health status from the fully adjusted models in the sensitivity analyses, we agree with the reviewers concern regarding the potential mediator effect of self-reported health status in our study. This removal did not result in substantial changes in the estimates, whether for education level, household composition or housing conditions, and thus, does not hamper the validity of the obtained results in the main models. While for some cancer types there was a tendency towards a slightly more pronounced association between the distinct socioeconomic or -demographic parameters and outcome, when removing self-reported health from the models, this relationship was not consistently observed for all cancer types.

Taken together, we provided crude, fully adjusted and partially adjusted estimates in the manuscript. To facilitate a correct interpretation of our results by the reader, we additionally added some reflections regarding potential mediator roles of certain parameters in the discussion section.

Text added in the Discussion section (lines 330-338):

“Through sensitivity analyses, we evaluate the potential mediating effect of self-reported health status [75-78]. While for some cancer types there was a tendency towards a slightly more pronounced association between the distinct SE or SD parameters and outcome, when removing self-reported health from the models, this finding was not consistently observed for all studied cancer types. Further, including all SE or SD parameters into one model, the aim of our study was to single-out the association between each indicator and the outcome. Crude estimates for each SE or SD parameter only adjusted for age were also calculated, resulting in slightly more pronounced associations compared to those obtained in the fully adjusted models. However, this did not result in substantial changes in IRR and OR estimates”.

Text added in the Results section – 2.1. Socioeconomic position and cancer incidence (line 140):

“This was particularly true for lung and head and neck cancers”.

 Text added in the Results section – 2.2. Socioeconomic position and stage distribution (lines 183-187):

“Estimates from the crude models for stage availability and stage at diagnosis, including only one SE or SD parameter per model and only adjusted for age, differed only slightly compared to those obtained in the full adjusted models. The association between SE or SD parameters and stage at diagnosis in the crude models was slightly more pronounced for head and neck cancers and malignant melanoma compared to the full adjusted models”.

Text added in the Results section – 2.3. Sensitivity analyses (lines 195-198):

“While for some cancer types there was a tendency towards a slightly more pronounced association between the distinct SE or SD parameters and outcome, this relationship was not consistently observed for all cancer types”.

References added:

Jackson, S. E.;  Williams, K.;  Beeken, R. J.; Steptoe, A., Changes in Health and Wellbeing in the Years Leading up to a Cancer Diagnosis: A Prospective Cohort Study. Cancer Prev Res (Phila) 2019, 12 (2), 79-88.

Manderbacka, K., Examining what self-rated health question is understood to mean by respondents. Scand J Soc Med 1998, 26 (2), 145-53.

Roelsgaard, I. K.;  Olesen, A. M.;  Simonsen, M. K.; Johansen, C., Self-rated health and cancer risk - a prospective cohort study among Danish women. Acta Oncol 2016, 55 (9-10), 1204-1209.

Wennberg, P.;  Rolandsson, O.;  Jerden, L.;  Boeing, H.;  Sluik, D.;  Kaaks, R.;  Teucher, B.;  Spijkerman, A.;  Bueno de Mesquita, B.;  Dethlefsen, C.;  Nilsson, P.; Nothlings, U., Self-rated health and mortality in individuals with diabetes mellitus: prospective cohort study. BMJ Open 2012, 2 (1), e000760.

A general comment: The concept “socioeconomic position” is recommend instead of “socioeconomic status”, consider to revise in the entire manuscript (Galobardes B, Shaw M, Lawlor DA, Lynch JW, Davey Smith G. Indicators of socioeconomic position (part 1). J Epidemiol Community Health. 2006 Jan;60(1):7-12. doi: 10.1136/jech.2004.023531. PMID: 16361448; PMCID: PMC2465546.) And please revise the manuscript once more for the different use of concepts: Socioeconomic status, socioeconomic deprivation ect.

Socioeconomic status (SES) has been changed into socioeconomic position (SEP) throughout the manuscript, including the title, as suggested by the reviewer. The term socioeconomic deprivation was removed from the manuscript. 

L.18: Revise to: “whether socioeconomic position is associated with stage availability and distribution”?

Lines 18-20: “Cancer stage being a major determinant for prognosis, a second aim of this study was to explore whether socioeconomic position is associated with stage availability and distribution”.

L22: Revise to: “socioeconomic position and cancer incidence”

Lines 20-24: “This study, encompassing almost seven million individuals, identifies population groups at increased risk of cancer and unknown or advanced stage at diagnosis in Belgium, and contributes to building a more comprehensive picture of the complex and multifaceted nature of socioeconomic position and cancer incidence in Belgium”.

L31: Delete “more”

Lines 33-35: “Results: Deprived groups showed higher risks for lung and head and neck cancers, whereas an inverse relation was observed for malignant melanoma and female breast cancer”.

L49, l. 58: revise deprivation to position.

Lines 50-52: “However, the direction and magnitude of this dynamic and multifaceted relationship differs across cancer types, and the mechanisms by which SEP impacts cancer risk are multiple, diverse and interconnected [4-6]”.

Lines 61-62: “In Belgium, so far, most studies have focused on the impact of socioeconomic position on cancer mortality [16-19]”.

L61 consider to add.: Further, the inequlity in survival has increased over time. (Dalton SO, Olsen MH, Johansen C, Olsen JH, Andersen KK. Socioeconomic inequality in cancer survival - changes over time. A population-based study, Denmark, 1987-2013. Acta Oncol. 2019 May;58(5):737-744. doi: 10.1080/0284186X.2019.1566772. Epub 2019 Feb 11. PMID: 30741062.)

Text added (lines 64-65): “Further, Dalton et al. demonstrated that the inequality in cancer survival has increased over time [22]”.

Reference added: Dalton, S. O.;  Olsen, M. H.;  Johansen, C.;  Olsen, J. H.; Andersen, K. K., Socioeconomic inequality in cancer survival - changes over time. A population-based study, Denmark, 1987-2013. Acta Oncol 2019, 58 (5), 737-744.

Reviewer 2 Report

the review/revision has improved the manuscript and have no other comments on this useful paper.

Author Response

Comments and Suggestions Reviewer 2

the review/revision has improved the manuscript and have no other comments on this useful paper.

No modifications required.

Reviewer 3 Report

I would like to thank the authors of addressing most of the feedback. I still doubt the novelty of the present findings and their impact for the general readership of cancers.

Author Response

Comments and Suggestions Reviewer 3

I would like to thank the authors of addressing most of the feedback. I still doubt the novelty of the present findings and their impact for the general readership of cancers.

No modifications required.

This manuscript is a resubmission of an earlier submission. The following is a list of the peer review reports and author responses from that submission.

Round 1

Reviewer 1 Report

The manuscript by Rosskamp et al. addresses a highly relevant issue: socioeconomic inequality in cancer. An association between socioeconomic position (SEP) and cancer incidence and stage at diagnosis is well known and has been observed in numerous studies from various countries. According to the author this study is, however, the first nationwide study examining socioeconomic inequality in cancer incidence and stage at diagnosis in Belgium using individual-level indicators for SEP. Thus, the study contributes towards a more comprehensive overview of socioeconomic inequality in Belgium for the six different cancer types studied. The manuscript is well-written with logical reasoning.

Methods: Uniquely they have access to nationwide information on self-reported health, however, this indicator is a mediator and should not be adjusted for in the main analysis. This will, most likely, underestimate the total effect of the relationship of interest: SEP --> cancer incidence and  SEP --> cancer stage. It could, however, be interesting to extend the analyses by investigating the indirect effects of self-reported health on these associations.

Also, please investigate and state how handling of missing education in one category could bias the result.

Generel comments:

There should be a more consistent use of the concept “socioeconomic status”. Often social deprivation is used instead. As we see a socioeconomic gradient for the entire population, the issue is not only about social deprivation.

Specific comments:

L13+24: improve phrasing: “is not equal” e.g. differs across

L19: delete or rephrase “permitted to”

L44-45: rephrase in line with the causal direction: e.g. “between socioeconomic status and cancer occurrence”

L50: “Behavioural and contextual determinants” does not need to “be taken into account”, but the underlying causes behind socioeconomic inequality is both individual/behavioural factors and contextual factors.

L45+L49+L57: The issue it’s not only about social deprivation. It concerns the entire population, as we see a socioeconomic gradient. Use e.g. socioeconomic status.

L45, L48,L56: social --> socioeconomic

L65: rephrase in line with the causal direction: whether socioeconomic status are associated with stage availablility….

L74-75: consider to delete, and refere to Table 1 at the end of the paragraf.

L79-80: consider to delete, and refere to Table 2a in line 81 and Table 2b in line 82.

L205: include differential exposure vs. differential susceptibility in the argumentation

L208: revise sentence

L211: Discuss also the impact of HPV-status

L212: probably also because of e.g. differences in health care seeking behavior,

L213: relate the argumentation to common risk-factors

L228: rephrase “positive gradient”

L236: delete however

L283: how does inclusion of missing information as a distinct category bias the result?

L297: Exposure is self-reported, does that introduce bias, and in which direction?

328: self-reported health and probably also region of residence is a mediator and not a confounding variable that should be adjusted for.

330: include a discussion of the validity of the information on stage.

Author Response

RE:[Cancers] Manuscript ID: cancers-1052623 – Major Revisions

Date: January 29th 2020

Dear Guest Editor,

Dear Prof. Akinyemiju,

Thank you for providing us with the referee’s comments of the manuscript entitled “Socioeconomic status, cancer incidence and stage at diagnosis: a nationwide cohort study in Belgium”, which we submitted to your journal. We were pleased to read that you consider the manuscript potentially acceptable, provided revision. We thank the reviewers for their valuable remarks and suggestions, which helped us to improve our manuscript.

In the revised version of our manuscript, we thoroughly amended the text, taking into account the reviewers’ comments and suggestions. The content has considerably been reviewed and adapted to improve the readers’ understanding.

Below we provide you with a point-by-point answer to the suggestions raised by the reviewers (in italics).

The revised manuscript was submitted with a highlighted copy, which shows the modifications that have been made.

Thank you once again for considering this manuscript for publication in Cancers.

We look forward to your final decision.

Kind regards,

Michael Rosskamp, on behalf of all co-authors.

Comments and Suggestions for Authors: Reviewer 1

The manuscript by Rosskamp et al. addresses a highly relevant issue: socioeconomic inequality in cancer. An association between socioeconomic position (SEP) and cancer incidence and stage at diagnosis is well known and has been observed in numerous studies from various countries. According to the author this study is, however, the first nationwide study examining socioeconomic inequality in cancer incidence and stage at diagnosis in Belgium using individual-level indicators for SEP. Thus, the study contributes towards a more comprehensive overview of socioeconomic inequality in Belgium for the six different cancer types studied. The manuscript is well-written with logical reasoning.

Methods: Uniquely they have access to nationwide information on self-reported health, however, this indicator is a mediator and should not be adjusted for in the main analysis. This will, most likely, underestimate the total effect of the relationship of interest: SEP --> cancer incidence and  SEP --> cancer stage. It could, however, be interesting to extend the analyses by investigating the indirect effects of self-reported health on these associations.

We acknowledge that self-reported health and region of residence (see comment L328 on page 5 of this document) can be considered as potential mediators. Therefore, we performed sensitivity analyses, in which we omitted self-reported health status and region of residence as covariates. The estimates obtained from these sensitivity analyses can be found in Supplementary Tables S1 (cancer incidence), S3 (stage availability) and S4 (stage distribution). In most, but not all analyses, there was a tendency towards a slightly more pronounced association between SES and outcome. However, estimates obtained from the sensitivity analyses were not meaningfully different from those of the main analyses.

Text added in the results section (lines 185-191):

“2.3. Sensitivity analyses

To assess potential mediator effects, we omitted self-reported health status and region of residence as covariates, which did not result in substantial changes in IRR estimates (Supplementary Table S1).

Similar observations were made for stage availability and distribution: no substantial effect was observed when removing self-reported health and region of residence as adjustment factors (Supplementary Tables S3 and S4).”

Text added in the material and methods section (lines 404-408):

“4.3 Sensitivity analyses

To assess the potential mediator effects of self-reported health and region of residence on cancer incidence, stage availability and stage at diagnosis, adjusted estimates were calculated without considering those adjustments factors“.

Also, please investigate and state how handling of missing education in one category could bias the result.

We acknowledge that missing data can lead to potential bias in parameter estimation, and perhaps weakened generalizability of findings. However, the focus of the study is to evaluate patterns of associations between SES and outcomes (incidence, stage availability, stage distribution), and the impact of a potential bias by missing data is unlikely to be large enough to substantially impact patterns of associations.

Notwithstanding, by creating an additional, fictive category for missing values, the association between the variable of interest (i.e. education) and the outcome (cancer incidence or stage at diagnosis) might be underestimated or biased. However, by considering missing data as a distinct category, no assumption is made on the missingness, and the missing values are not replaced with a more or less reliable estimate.

Missing data is a complicated topic and can depend on the variable itself, on other variables (e.g. age) and on the outcome. In order to provide valid estimates, high knowledge with regards to multiple imputation methods along with sufficient and accurate data is required. Multiple imputation analysis would have been beyond the scope of this study, given the extensive number of analyses. In addition, replacing missing values with an estimate (using multiple imputation) may not solve the bias completely, whether missingness is randomly distributed or not. Furthermore, the focus of this paper is to evaluate patterns rather than the size of associations for different cancer types and SE factors. It is unlikely that a potential bias induced by missing data would influence the patterns of associations between cancer types.

Information on education level, housing conditions and self-reported health was missing in 12.8%, 8.5% and 5.4% of the study cohort, respectively.

In the present study, estimates obtained for missing education tended towards those observed for the lowest education category (i.e. primary or less) for some cancer types, suggesting that missingness is not randomly distributed, while for missing housing conditions no clear pattern was observed. Therefore, a complete-case only analysis would not have been an accurate alternative. Similarly, socioeconomic inequalities in health survey participation have already been demonstrated in the past.

Text added in the discussion section (lines 321-330):

“Missing data may have biased our results [76]. By creating an additional, fictive category for missing values, we included those observations in our study cohort, while not making any assumptions on the missingness of the data. Information on education level, housing conditions and self-reported health was missing in 12.8%, 8.5% and 5.4% of the study cohort, respectively. The estimates obtained for the category of missing education tended towards those observed for the lowest education category for some cancer types, suggesting that missingness is not randomly distributed, while for missing housing conditions no clear pattern was observed. However, the focus of this study was to evaluate patterns of association between SES and outcomes, and the impact of potential bias by missing data is unlikely to be large enough to substantially impact the observed patterns of association”.

Reference added:

Dong, Y.; Peng, C. Y., Principled missing data methods for researchers. Springerplus 2013, 2 (1), 222.

General comments:

There should be a more consistent use of the concept “socioeconomic status”. Often social deprivation is used instead. As we see a socioeconomic gradient for the entire population, the issue is not only about social deprivation.

The terms “socioeconomic status”, “socioeconomic deprivation”, “socioeconomic gradient” and “socioeconomic inequality” are now used in a more consistent way throughout the text.

Specific comments:

L13+24: improve phrasing: “is not equal” e.g. differs across

Line 13: “However, this relationship differs across cancer types and socioeconomic parameters”.

Lines 23-25: “Socioeconomic status is associated with cancer incidence, but the direction and magnitude of this relationship differs across cancer types, geographical regions and socioeconomic parameters”.

L19: delete or rephrase “permitted to”

Lines 19-22: “This study, encompassing almost seven million individuals, identifies population groups at increased risk of cancer and unknown or advanced stage at diagnosis in Belgium, and contributes to building a more comprehensive picture of the complex and multifaceted nature of socioeconomic deprivation and cancer incidence in Belgium”.

L44-45: rephrase in line with the causal direction: e.g. “between socioeconomic status and cancer occurrence”

Lines 44-47: “Several studies have shown a mostly consistent pattern between socioeconomic status (SES) and overall cancer occurrence, reflecting differences in the exposure to main risk factors and inequalities in access to prevention and early detection measures, with a consequent impact on survival and quality of life [2, 3]”.

L50: “Behavioural and contextual determinants” does not need to “be taken into account”, but the underlying causes behind socioeconomic inequality is both individual/behavioural factors and contextual factors.

Lines 49-52: “Behavioural and contextual determinants are underlying causes behind socioeconomic inequality and no single indicator can capture the complexity of socioeconomic (SE) and sociodemographic (SD) circumstances [7-9]”.

L45+L49+L57: The issue it’s not only about social deprivation. It concerns the entire population, as we see a socioeconomic gradient. Use e.g. socioeconomic status.                                                              L45, L48,L56: social --> socioeconomic

Lines 44-47: please see previous comment.

Lines 47-49: “However, the direction and magnitude of this dynamic and multifaceted relationship differs across cancer types, and the mechanisms by which socioeconomic deprivation impacts cancer risk are multiple, diverse and interconnected [4-6]”.

Lines 56-57: “However, stage availability and distribution do not seem to be equally dispersed among socioeconomic groups [12-15]”.

L65: rephrase in line with the causal direction: whether socioeconomic status are associated with stage availability….

Lines 65-67: “Second, this study also aims to explore whether socioeconomic status is associated with disparities in stage availability and distribution”.

L74-75: consider to delete, and refere to Table 1 at the end of the paragraf.

Lines 72-75: “During the study period (63,828,547.3 person-years), 280,019 (4.0%) individuals were diagnosed with at least one of the six studied cancer types (286,220 diagnoses), 1,020,999 (14.7%) died, 146,648 (2.1%) were lost to follow-up and 5,792,724 (83.2%) were still alive at the end of the observation period (December 31st 2013 ; Table 1)”.

Lines 75-76: deleted.

L79-80: consider to delete, and refere to Table 2a in line 81 and Table 2b in line 82.

Lines 80-81: deleted.

Lines 81-85: “The number of cancer diagnoses ranged from 7,311 malignant melanoma to 54,149 lung cancer cases in men (Table 2a), and from 4,338 head and neck cancer to 97,650 breast cancer cases in women (Table 2b). For most cancer types, truncated age-standardized incidence rates (ASR) were inversely associated with education and housing comfort”.

L205: include differential exposure vs. differential susceptibility in the argumentation

Lines 232-233: “Differences in exposure but also in susceptibility to common risk factors across socioeconomic groups may lead to the observed disparities in cancer incidence”. 

L208: revise sentence

Lines 227-228: “Further understanding in lung cancer aetiology is needed to clarify the pathways linking SES to lung cancer”.

L211: Discuss also the impact of HPV-status

Lines 230-232: “In addition, Krupar et al. showed that prevalence of HPV infection, an independent risk factor for different head and neck cancers, differed across regions and socioeconomic factors [37]”.

Reference added:

Krupar, R.;  Hartl, M.;  Wirsching, K.;  Dietmaier, W.;  Strutz, J.; Hofstaedter, F., Comparison of HPV prevalence in HNSCC patients with regard to regional and socioeconomic factors. Eur Arch Otorhinolaryngol 2014, 271 (6), 1737-45.

L212: probably also because of e.g. differences in health care seeking behavior,                           L213: relate the argumentation to common risk-factors

Lines 232-237: “Differences in exposure but also in susceptibility to common risk factors across socioeconomic groups may lead to the observed disparities in cancer incidence. Further, lower SES was associated with advanced stage at diagnosis for lung and head and neck cancers in men, which might be influenced by fatalistic beliefs and attitudes, differences in health care seeking behavior and a higher prevalence of comorbid conditions related to common risk factors, entailing possible misinterpretation of alarming symptoms and delay in presentation at the clinic [38-43]”.

L228: rephrase “positive gradient”

Lines 250-252: “These findings suggest that the association between level of education and breast cancer risk could, to a substantial degree, be explained by these established risk factors [51-54]”.

L236: delete however

Lines 258-259: “The relationship between colorectal cancer and SES is not homogeneous between European countries”.

L283: how does inclusion of missing information as a distinct category bias the result?

Please see second comment (page 2 of this document).

L297: Exposure is self-reported, does that introduce bias, and in which direction?

Exposure being self-reported, results might be affected, amongst others, from subjective perception and social desirability. However, SE and SD parameters being clearly defined through categorical variables, we do not expect large impact of potential bias due to self-reported exposure. More objective measures of exposure  (e.g. the administrative census of 2011) could help to assess the potential bias of self-reported exposure and its association with outcome at population level in future.

Text added in the discussion section (lines 309-311):

“Further, self-reported exposure data can potentially affect results through subjective perception or social desirability, amongst others [72].”

Reference added:

Althubaiti, A., Information bias in health research: definition, pitfalls, and adjustment methods. J Multidiscip Healthc 2016, 9, 211-7.

L328: self-reported health and probably also region of residence is a mediator and not a confounding variable that should be adjusted for.

Please see first comment (page 1 of this document).

L330: include a discussion of the validity of the information on stage.

Text added in the material and methods section (lines 361-367):

“Validity of stage information depends strongly on the data received from the sources, i.e. the oncological care programs and the laboratories for pathological anatomy. All data that enters the BCR is submitted to an extended set of automated and manual validation procedures based on IARC (International Agency for Research on Cancer) guidelines to ensure validity and quality of the data. The data source is consulted to provide additional details for cases with an uncertain diagnosis, insufficient, erroneous or conflicting information“.

Reviewer 2 Report

Thanks for this interesting study from Belgium. I support publication as this is an understudied subject in the context of the national population in Belgium and the study has important findings. However, a number of improvements are needed regarding the methodology and its reporting.

  1. For the Poisson models (incidence) and the logistic models, it is not clear whether data presented (in the main text and figures) are adjusted or crude. Good practice (e.g. STROBE or RECORD) guidelines is to estimate and report / interpret both crude (observed) and adjusted findings. As the findings are presented currently, we do not know if there is an apparent gradient for educational status category that is much attenuated once adjustment is made for other variables, or whether why we see is indeed the adjusted / net effect. Similar question for all other variables. This is a crucial limitation of the paper as currently written and presented. I would urge the authors not only to report both the crude and adjusted models but to also interpret the degree of differences between coefficients from the crude and the adjusted models. The methods section needs to be also very clear as to the design / construction of the Poisson and the logistic models (what were they adjusted to as exposure variables).
  2. For the advanced stage analysis (not the incidence) please also report tables with observed %. Currently you over-emphasise significant differences (as judged by p value) lumping together effect sizes that are substantially different in terms of size/strength (e.g. an effect size of 1.1 is pretty marginally important really, whereas one of 1.5 is). Please be more nuanced and discriminant in your interpretations and do not 'call' all such heterogeneous effects the same way.
  3. In Discussion, please acknowledge / discuss limitation regarding missing stage data, in principle those could be imputed if you have information on survival (though you may not have had such data) - please discuss the issue. (I appreciate that % complete stage was quite high overall, and that you have appropriately reported variation in % missing by variable category, so these are good features of your study).
  4. In general the theoretical framing as to explaining potential variation in advanced stage is a bit limited, please consider adding references to fatalism, particularly for lung cancer, e.g. reference of possible use is https://pubmed.ncbi.nlm.nih.gov/25650183/ . Also, regarding your hypothesis that morbidity may be a confounder of socioeconomic variation in stage, please consider citing this review paper that includes all plausible mechanisms by which morbidity may influence stage at diagnosis including the alternative explanations hypothesis that you propose https://pubmed.ncbi.nlm.nih.gov/31350467/ 
  5. Similarly, there is little in the way of theoretical discussion / explanation of the fact that the biggest socio-economic differences in stage at diagnosis relate to the two 'easier-to-suspect' cancers post-presentation of those examined (melanoma and breast cancer) - pointing to potential delays in help-seeking being the issue, i.e. patient behaviour / patient literacy factors. You may wish to consider these papers as of relevance https://pubmed.ncbi.nlm.nih.gov/25734380/ also https://pubmed.ncbi.nlm.nih.gov/25491791/ . Basically, there is very little potential delay in diagnosis once patients have presented with a breast lump or a mole/suspected melanomatous lesion, as those 2 cancers are 'very easy to suspect' https://pubmed.ncbi.nlm.nih.gov/22365494/ , and whatever diagnostic delay there may be it will not be differential between different social groups to the length required to generate tumour stage progression, therefore the differences are likely reflecting delayed presentation (or at least the is a potential theoretical explanation that needs to be considered among the others).

I hope above useful and look forward to seeing your papers progress to publication.

Author Response

RE:[Cancers] Manuscript ID: cancers-1052623 – Major Revisions

Date: January 29th 2020

Dear Guest Editor,

Dear Prof. Akinyemiju,

Thank you for providing us with the referee’s comments of the manuscript entitled “Socioeconomic status, cancer incidence and stage at diagnosis: a nationwide cohort study in Belgium”, which we submitted to your journal. We were pleased to read that you consider the manuscript potentially acceptable, provided revision. We thank the reviewers for their valuable remarks and suggestions, which helped us to improve our manuscript.

In the revised version of our manuscript, we thoroughly amended the text, taking into account the reviewers’ comments and suggestions. The content has considerably been reviewed and adapted to improve the readers’ understanding.

Below we provide you with a point-by-point answer to the suggestions raised by the reviewers (in italics).

The revised manuscript was submitted with a highlighted copy, which shows the modifications that have been made.

Thank you once again for considering this manuscript for publication in Cancers.

We look forward to your final decision.

Kind regards,

Michael Rosskamp, on behalf of all co-authors.

Comments and Suggestions for Authors: Reviewer 2

Thanks for this interesting study from Belgium. I support publication as this is an understudied subject in the context of the national population in Belgium and the study has important findings. However, a number of improvements are needed regarding the methodology and its reporting.

  1. For the Poisson models (incidence) and the logistic models, it is not clear whether data presented (in the main text and figures) are adjusted or crude. Good practice (e.g. STROBE or RECORD) guidelines is to estimate and report / interpret both crude (observed) and adjusted findings. As the findings are presented currently, we do not know if there is an apparent gradient for educational status category that is much attenuated once adjustment is made for other variables, or whether why we see is indeed the adjusted / net effect. Similar question for all other variables. This is a crucial limitation of the paper as currently written and presented. I would urge the authors not only to report both the crude and adjusted models but to also interpret the degree of differences between coefficients from the crude and the adjusted models. The methods section needs to be also very clear as to the design / construction of the Poisson and the logistic models (what were they adjusted to as exposure variables).

Text added in the results section (lines 104-106):

“Incidence rate ratios (IRR) were adjusted for age, region of residence and self-reported health status in multivariable models. Adjusted IRR estimates by SE or SD parameter, cancer type and sex are shown in Figure 1 and Supplementary Table S1”.

Text added in the results section (lines 134-136):

“Crude incidence models, including one SE or SD parameter per model and only adjusted for age, resulted in slightly more pronounced associations compared to the estimates obtained in the full adjusted models (Supplementary Table S1)”.

Text added in the results section (lines 144-146):

“Both stage availability (Figure 2 and Supplementary Table S3) and stage at diagnosis (Figure 3 and Supplementary Table S4), reported as adjusted odds ratios (OR), were related to SE and SD parameters, but the magnitude of association differed largely by cancer type and sex”.

Text added in the results section (lines 178-180):  

“Estimates from the crude models for stage availability and stage at diagnosis, including only one SE or SD parameter per model and only adjusted for age, were slightly more pronounced compared to those obtained in the full adjusted models (Supplementary Tables S3 and S4)”.

Text added in the material and methods section (lines 379-386):

“Adjusted incidence rate ratios (IRR) were computed through Poisson regression models with the number of diagnoses as dependent variable. Education level, household composition and housing conditions were included in every model as independent variables and age at start of follow-up, region of residence and self-reported health at time of census as control variables. The log of the person-time was used as offset variable.

Crude estimates for the associations between SE or SD parameters and cancer incidence were assessed by including a single SE or SD parameter per model, and only adjusting for age, which was considered a major confounder”.

Text added in the material and methods section (lines 393-396):

“Education level, household composition and housing conditions were included in every model and adjustment factors included age and region of residence at time of diagnosis, as well as self-reported health at time of census.

The crude models included a single SE or SD parameter per model and were only adjusted for age, considered a major confounder”.

  1. For the advanced stage analysis (not the incidence) please also report tables with observed %. Currently you over-emphasise significant differences (as judged by p value) lumping together effect sizes that are substantially different in terms of size/strength (e.g. an effect size of 1.1 is pretty marginally important really, whereas one of 1.5 is). Please be more nuanced and discriminant in your interpretations and do not 'call' all such heterogeneous effects the same way.

Supplementary Table S2 provides the stage at diagnosis availability and distribution by cancer type and sex.

Text added in the results section (lines 138-141):

“Cancer stage was available in 86.4% of cases ranging from 92.6% in female breast cancer to 77.2% in men lung cancer (Supplementary Table S2). The proportion of advanced stage (stage III or IV) diagnoses varied substantially by cancer type: from 6.6% for malignant melanoma in women to 55.9% for lung cancer in men”.

Text added in the discussion section (lines 288-289):

“However, overall stage availability and distribution differed across cancer types. This should be considered when interpreting effect size which varied across cancer types.

  1. In Discussion, please acknowledge / discuss limitation regarding missing stage data, in principle those could be imputed if you have information on survival (though you may not have had such data) - please discuss the issue. (I appreciate that % complete stage was quite high overall, and that you have appropriately reported variation in % missing by variable category, so these are good features of your study).

Cancer stage is a key predictor of cancer survival and completeness of registration is crucial for understanding outcomes at population level. While cancer registries aim to collect detailed information on the extent of disease, stage information may be missing, introducing methodological difficulties for the analysis and interpretation of results. Whether missing stage information is randomly distribution amongst SE groups may be questioned, though.

Rather than imputing estimates for missing stage, we therefore aimed to evaluate stage availability as an outcome in our study and described the associations between SES and missing stage information for six different cancer types. While the proportion of patients for whom there was no valid stage information available differed across cancer types and age groups, we found differences in the association between the three studied socioeconomic parameters and cancer stage availability and discussed the potential underlying reasons of missing stage information by cancer type.

Missing stage information following a pattern rather than being randomly distributed, its impact on stage distribution analyses needs to be acknowledged.

Text added in the discussion section (lines 290-294):

“The observed disparities in stage availability likely have an impact on the differences found in stage distribution and results require careful interpretation. Stage distribution analyses were performed in a subgroup of the cancer population (i.e. those with a known stage) why generalization may not be feasible given that our analyses on stage availability indicated a socioeconomic gradient for missing stage”.

  1. In general the theoretical framing as to explaining potential variation in advanced stage is a bit limited, please consider adding references to fatalism, particularly for lung cancer, e.g. reference of possible use is https://pubmed.ncbi.nlm.nih.gov/25650183/. Also, regarding your hypothesis that morbidity may be a confounder of socioeconomic variation in stage, please consider citing this review paper that includes all plausible mechanisms by which morbidity may influence stage at diagnosis including the alternative explanations hypothesis that you propose https://pubmed.ncbi.nlm.nih.gov/31350467/

Text added in the discussion section (lines 233-237):

“Further, lower SES was associated with advanced stage at diagnosis for lung and head and neck cancers in men, which might be influenced by fatalistic beliefs and attitudes, differences in health care seeking behavior and a higher prevalence of comorbid conditions related to common risk factors, entailing possible misinterpretation of alarming symptoms and delay in presentation at the clinic [38-43]”.

References added:

Lyratzopoulos, G.;  Liu, M. P.;  Abel, G. A.;  Wardle, J.; Keating, N. L., The Association between Fatalistic Beliefs and Late Stage at Diagnosis of Lung and Colorectal Cancer. Cancer Epidemiol Biomarkers Prev 2015, 24 (4), 720-6.

Renzi, C.;  Kaushal, A.;  Emery, J.;  Hamilton, W.;  Neal, R. D.;  Rachet, B.;  Rubin, G.;  Singh, H.;  Walter, F. M.;  de Wit, N. J.; Lyratzopoulos, G., Comorbid chronic diseases and cancer diagnosis: disease-specific effects and underlying mechanisms. Nat Rev Clin Oncol 2019, 16 (12), 746-761.

  1. Similarly, there is little in the way of theoretical discussion / explanation of the fact that the biggest socio-economic differences in stage at diagnosis relate to the two 'easier-to-suspect' cancers post-presentation of those examined (melanoma and breast cancer) - pointing to potential delays in help-seeking being the issue, i.e. patient behaviour / patient literacy factors. You may wish to consider these papers as of relevance https://pubmed.ncbi.nlm.nih.gov/25734380/ also https://pubmed.ncbi.nlm.nih.gov/25491791/ . Basically, there is very little potential delay in diagnosis once patients have presented with a breast lump or a mole/suspected melanomatous lesion, as those 2 cancers are 'very easy to suspect' https://pubmed.ncbi.nlm.nih.gov/22365494/ , and whatever diagnostic delay there may be it will not be differential between different social groups to the length required to generate tumour stage progression, therefore the differences are likely reflecting delayed presentation (or at least the is a potential theoretical explanation that needs to be considered among the others).

Text added in discussion section (lines 265-272):

“Some cancer types are hard to suspect as patients present with non-specific symptoms (e.g. chest, back or abdominal pain). This diagnostic difficulty, potentially enforced by comorbid conditions of the patient, may lead to multiple consultations in primary care and delay in hospital referral, increasing the risk of stage progression. For other cancer types, specific signs and symptoms, such as palpable breast lumps or visible skin lesions, make cancer easier to suspect. In our study, differences in stage distribution across SE groups tended to be more pronounced for these cancer types and likely point towards potential delays in medical consultation and differences in help-seeking behaviour and health literacy of the patients [61-63]”.

References added:

Lyratzopoulos, G.;  Neal, R. D.;  Barbiere, J. M.;  Rubin, G. P.; Abel, G. A., Variation in number of general practitioner consultations before hospital referral for cancer: findings from the 2010 National Cancer Patient Experience Survey in England. Lancet Oncol 2012, 13 (4), 353-65.

Lyratzopoulos, G.;  Wardle, J.; Rubin, G., Rethinking diagnostic delay in cancer: how difficult is the diagnosis? BMJ 2014, 349, g7400.

Lyratzopoulos, G.;  Saunders, C. L.;  Abel, G. A.;  McPhail, S.;  Neal, R. D.;  Wardle, J.; Rubin, G. P., The relative length of the patient and the primary care interval in patients with 28 common and rarer cancers. Br J Cancer 2015, 112 Suppl 1, S35-40.

I hope above useful and look forward to seeing your papers progress to publication.

Reviewer 3 Report

With interest I read the manuscript “Socioeconomic status, cancer incidence and stage at 2 diagnosis: a nationwide cohort study in Belgium”. However I have a few concerns before publications. My main concern is the lack of novelty. However, the following concerns also need to be addressed:

-- Major and minor comments --

  1. This study lacked the information regarding risk factors for cancer. Important confounders miss in the analysis.
  2. There are many tables and figures and the authors should select only the important ones. The result section is quite long.
  3. The discussion illustrates similarities with previous literature but stays short in discussing the novelty of the present findings.

Author Response

RE:[Cancers] Manuscript ID: cancers-1052623 – Major Revisions

Date: January 29th 2020

Dear Guest Editor,

Dear Prof. Akinyemiju,

Thank you for providing us with the referee’s comments of the manuscript entitled “Socioeconomic status, cancer incidence and stage at diagnosis: a nationwide cohort study in Belgium”, which we submitted to your journal. We were pleased to read that you consider the manuscript potentially acceptable, provided revision. We thank the reviewers for their valuable remarks and suggestions, which helped us to improve our manuscript.

In the revised version of our manuscript, we thoroughly amended the text, taking into account the reviewers’ comments and suggestions. The content has considerably been reviewed and adapted to improve the readers’ understanding.

Below we provide you with a point-by-point answer to the suggestions raised by the reviewers (in italics).

The revised manuscript was submitted with a highlighted copy, which shows the modifications that have been made.

Thank you once again for considering this manuscript for publication in Cancers.

We look forward to your final decision.

Kind regards,

Michael Rosskamp, on behalf of all co-authors.

Comments and Suggestions for Authors: Reviewer 3

With interest I read the manuscript “Socioeconomic status, cancer incidence and stage at 2 diagnosis: a nationwide cohort study in Belgium”. However I have a few concerns before publications. My main concern is the lack of novelty. However, the following concerns also need to be addressed:

 -- Major and minor comments --

1. This study lacked the information regarding risk factors for cancer. Important confounders miss in the analysis.

Presenting results of a population-based observational study, we acknowledge the lack of information on some important risk factors and residual confounding cannot be excluded.

Text added in the discussion section (lines 304-307):

“We could not account for known risk factors (i.e. alcohol and tobacco consumption, Body Mass Index, HPV status in head and neck cancers), reproductive history (breast cancer), nor participation in screening programs (breast and colorectal cancer) and residual confounding cannot be excluded.”

2. There are many tables and figures and the authors should select only the important ones. The result section is quite long.

We acknowledge that the result section can be considered to be quite long. However, we received comments from the two other reviewers to further extent the results. To accommodate the comment of Reviewer 3, we tried to place results that are not essential but informative as Online Supplement and hope the Reviewer understands our reasoning.

In order to address the comments and suggestions made by Reviewers 1&2, we extended the Supplementary Tables 1 (cancer incidence), 3 (stage availability) and 4 (stage distribution) to provide crude (single SE parameter + age) model estimates and those obtained when removing self-reported health and region of residence (potential mediators) as adjustment factors. Amendments were made in the material and methods and results section including a paragraph on sensitivity analyses and an additional Supplementary Table (2) was added showing differences in stage availability and distribution by cancer type and sex.   

3. The discussion illustrates similarities with previous literature but stays short in discussing the novelty of the present findings.

Text added in the discussion section (lines 195-199):

“Adding to the growing body of evidence, it also provides new and important findings in this yet understudied subject on a nationwide scale in the light of the Belgian health care system. The study contributes towards a more comprehensive overview of socioeconomic inequality in cancer burden in Belgium, a country known for its obliged and advanced health insurance”.